# Gengricin^®^: A Nutraceutical Formulation for Appetite Control and Therapeutic Weight Management in Adults Who Are Overweight/Obese

**DOI:** 10.3390/ijms25052596

**Published:** 2024-02-23

**Authors:** Elisabetta Schiano, Fortuna Iannuzzo, Mariano Stornaiuolo, Fabrizia Guerra, Gian Carlo Tenore, Ettore Novellino

**Affiliations:** 1Inventia Biotech—Healthcare Food Research Center s.r.l., Strada Statale Sannitica KM 20.700, 81020 Caserta, Italy; elisabettaschiano@inventiabiotech.com; 2Department of Pharmacy, University of Chieti-Pescara G. D’Annunzio, 66100 Chieti, Italy; fortuna.iannuzzo@unich.it; 3Department of Pharmacy, University of Naples Federico II, Via Domenico Montesano 59, 80131 Naples, Italy; mariano.stornaiuolo@unina.it (M.S.); giancarlo.tenore@unina.it (G.C.T.); 4NGN Healthcare—New Generation Nutraceuticals s.r.l., Torrette Via Nazionale 207, 83013 Mercogliano, Italy; ngnhealthcare@gmail.com; 5Department of Medicine and Surgery, Catholic University of the Sacred Heart, 00168 Rome, Italy

**Keywords:** nutraceuticals, TAS2R, bitter taste, gut hormones, obesity, satiety control

## Abstract

In the field of nutritional science and metabolic disorders, there is a growing interest in natural bitter compounds capable of interacting with bitter taste receptors (TAS2Rs) useful for obesity management and satiety control. This study aimed to evaluate the effect of a nutraceutical formulation containing a combination of molecules appropriately designed to simultaneously target and stimulate these receptors. Specifically, the effect on CCK release exerted by a multi-component nutraceutical formulation (*Cinchona* bark, *Chicory*, and *Gentian* roots in a 1:1:1 ratio, named Gengricin^®^) was investigated in a CaCo-2 cell line, in comparison with *Cinchona* alone. In addition, these nutraceutical formulations were tested through a 3-month randomized controlled trial (RCT) conducted in subjects who were overweight–obese following a hypocaloric diet. Interestingly, the Gengricin^®^ group exhibited a significant greater weight loss and improvement in body composition than the Placebo and *Cinchona* groups, indicating its effectiveness in promoting weight regulation. Additionally, the Gengricin^®^ group reported higher satiety levels and a significant increase in serum CCK levels, suggesting a physiological basis for the observed effects on appetite control. Overall, these findings highlight the potential of natural nutraceutical strategies based on the combination of bitter compounds in modulating gut hormone release for effective appetite control and weight management.

## 1. Introduction

Obesity, a complex and multifactorial condition characterized by excessive adiposity or body fat, currently affects over a third of the global population [1]. It represents a significant risk factor for numerous pathologies, including cardiovascular diseases, type 2 diabetes, and cancers. Currently, obesity has reached epidemic proportions worldwide, affecting both adults and children, resulting in a substantial financial burden on healthcare systems. The World Health Organization (WHO) defines obesity as an excessive accumulation of body fat, diagnosed by a body mass index (BMI) ≥ 30 kg/m^2^, which may affect health [2,3,4]. In this context, there is a common interest in finding successful treatments to counteract or prevent obesity and promote individual health. While recommendations for obesity management represent a cornerstone in clinical practice, their effective implementation often encounters a significant challenge: low patient compliance. This phenomenon can be attributed to various factors ranging from the complexity of implementing recommendations to the chronic and multifactorial nature of obesity itself [5]. Guidelines for obesity treatment typically encompass lifestyle modifications, such as adopting a balanced diet and increasing physical activity, alongside potential pharmacological therapies and surgical interventions in select cases [6]. However, translating these directives into practice necessitates ongoing commitment and substantial willpower from patients. The absence of immediate and visible results, common in many obesity management programs, can further diminish patient motivation. Additionally, psychological issues (e.g., stress, depression, and anxiety), socioeconomic barriers, and limited access to specialized medical care can negatively influence compliance, presenting additional challenges [7].

Nowadays, natural compounds derived from food sources demonstrate their efficacy in numerous pathologies, emerging as a safer alternative to conventional-approach therapy, which often involves a combination of drugs, often associated with various side effects [8,9]. Despite the growing interest in nutraceuticals as potential tools for obesity prevention and management, the understanding of their efficacy and safety remains limited. The field of nutraceutical research in relation to obesity is still in its early stages and there is a need for comprehensive studies on their mechanisms of action, optimal dosages, and long-term effects [10]. This knowledge gap underscores the necessity for further rigorous clinical trials and mechanistic studies to establish evidence-based guidelines for the utilization of nutraceuticals in the multifaceted battle against obesity.

In this scenario, there is a growing interest in bioactive compounds capable of interacting with bitter taste receptors (TAS2Rs) useful for obesity management and satiety control [11]. TAS2Rs, a family of G-protein-coupled receptors, are known for their ability to detect bitter compounds. These receptors are not only located in the gustatory system but are also expressed in various extra-oral tissues, including the gastrointestinal tract, where they play a significant role in mediating metabolic processes [12]. The activation of TAS2Rs in the stomach and duodenum has been shown to trigger a cascade of hormonal responses critical for appetite regulation. For instance, the stimulation of TAS2R in the gastric mucosa leads to the release of ghrelin, an orexigenic hormone that signals hunger to the brain [13]. Conversely, TAS2R activation in the duodenal lining slows down duodenal motility and promotes the release of cholecystokinin (CCK-B), a peptide known to induce satiety by delaying gastric emptying and promoting a sense of fullness [14,15]. This suggests a potential avenue for exploring natural compounds that target these receptors to influence satiety signals, potentially offering a novel approach in the pursuit of effective strategies for weight management and obesity prevention. Furthermore, our previous studies identified melatonin as a likely endogenous mediator of TAS2Rs’ activity. Melatonin, widely distributed throughout the central nervous system and gastrointestinal tract, may act as a natural ligand for duodenal TAS2Rs, thereby influencing the receptor’s activity and its downstream effects on appetite and satiety signals [16]. More specifically, at the stomach level, melatonin exerts its effects by binding to inhibitory regulative G-protein-coupled receptors, which reduce ghrelin secretion, a hormone primarily associated with hunger signaling. Moving along the digestive tract to the duodenum, melatonin engages with a different subtype of receptors, specifically the Gq-coupled proteins. The activation of the latter by melatonin increases the intracellular calcium within the duodenal cells, leading to the secretion of CCK, a hormone integral to the digestion process and satiety sensation [16,17]. Therefore, with the aim of identifying natural-derived substances that share a pharmacophoric portion with melatonin, a similarity screening search was carried out. This screening revealed that the alkaloids from *Cinchona Succirubra* (quinine, quinidine, cinchonine, and cinchonidine) all have a suitably functionalized ring structure that overlaps with the pharmacophore of melatonin [16,18]. In support of the potential role of alkaloids from *Cinchona* as modulators of the TAS2R, a very recent study investigated the effects of *Cinchona* supplementation on weight loss, appetite, and serum cholecystokinin (CCK) levels in adults who are overweight and obese following a hypocaloric diet. The results showed a significant reduction in the body weight (from 96.9 ± 1.6 kg to 90.4 ± 1.7 kg, *p* < 0.0001) and percentage of fat mass (from 34.6 ± 1.7% to 31.1 ± 2.1%, *p* < 0.0001, in the treatment group, compared to Placebo group). In addition, these subjects maintained higher serum CCK levels, which were associated with increased satiety over the 60-day treatment period, compared to the Placebo group [19]. Overall, these results indicated that *Cinchona* supplementation, in concomitance with a hypocaloric diet, effectively supported weight management and appetite control by influencing CCK levels.

In the realm of gustatory perception, the interaction of specific bitter compounds with TAS2Rs presents a complex and intriguing biochemical puzzle. Multiple TAS2Rs co-locate within the same taste receptor cell (TRC), inducing a distinct intracellular response that is independent of the specific type of bitter tasting involved [16,20]. Quinine alkaloids, with their pronounced bitter profile related to their structural complexity and potency, are reported to interact with a distinct set of TAS2Rs (EC_50_ value of 10 µM) such as TAS2R4, TAS2R7, TAS2R10, TAS239, and TAS2R46 [21,22]. Similarly, the bitter constituents of *Gentian*, like gentiopicrin and amarogentin, demonstrate a unique receptor-binding pattern (EC_50_ values of 30, 10, 3, and 100 µM for TAS2R43, TAS2R46, TAS2R47, and TAS2R50 receptors, respectively) [20,22,23]. This distinct molecular interaction is reflective of its unique molecular architecture, distinguished by their glycoside nature [24]. In contrast, lactucin, lactucopicrin, and 11β,13-dihydrolactucin from *Chicory* show high affinity toward TAS2Rs (lactupicrin EC_50_ values of 0.46 and 0.35 µM for TAS2R43 and TAS2R46, respectively), aligning with their simpler lactone structures [25]. The structural complexity of these compounds, ranging from quinine’s complex alkaloid structure to the relatively simpler lactone structures of lactucin and lactucopicrin, suggests a nuanced interaction with these receptor subtypes, suggesting a potential for both distinct and overlapping receptor activation. Interestingly, the diverse receptor activation profiles of these compounds raise the possibility of synergistic effects on the overall activation of bitter taste receptors, as the potential to activate multiple TAS2Rs is influenced by the quantity and combination of “functional groups” within bitter substances, or by the ligand’s ability to create a “common bitter motif” that associates with low affinity in the binding sites of various TAS2Rs [22,26]. Therefore, the investigation of this potential synergism could provide deeper insights into the multifaceted nature of bitterness and its perception, driven by the interaction of different bitter compounds with a varied array of TAS2Rs.

Based on these observations and considering that each receptor possesses unique specificity and affinity for different classes of bitter compounds, we herein aimed to evaluate the effect of a nutraceutical product containing a combination of molecules appropriately designed to simultaneously target and stimulate these receptors. Specifically, the effect on CCK release exerted by a multi-component nutraceutical formulation (*Cinchona* bark, *Chicory* root, and *Gentian* root in a 1:1:1 ratio, named Gengricin^®^) was investigated in a CaCo-2 cell line, in comparison with the *Cinchona* extract alone. In addition, the above-mentioned nutraceutical formulations were tested for their effects on satiety, weight loss, and body composition changes through a randomized controlled trial (RCT) conducted in subjects who were overweight–obese following a hypocaloric diet.

## 2. Results

### 2.1. Cholecystokinin Release in CaCo-2 Cells

To test the different response in terms of CKK secretion, CaCo-2 cells underwent pre-incubation either with the *Cinchona* extract alone or with the multi-component nutraceutical formulation, named Gengricin^®^. As reported in Figure 1, although cells were demonstrated to increase the CCK release upon treatment with both treating samples (*p* < 0.0001 vs. control), the incubation with Gengricin^®^ was able to induce a significantly higher CCK secretion (+15% vs. *Cinchona* bark extract, *p* < 0.05). Therefore, this observed effect underscores a likely synergistic interplay among the components of the multi-ingredient nutraceutical formulation on the overall activation of different bitter taste receptors.

### 2.2. Randomized Controlled Trial (RCT)

#### 2.2.1. Evaluation of Demographic and Biochemical Parameters

A total of 92 individuals (Placebo group, *n* = 31; *Cinchona* group, *n* = 30; Gengricin^®^ group, *n* = 31) successfully completed the three-month intervention. Data concerning general and biochemical characteristics of the patients at baseline and after three months are presented in Table 1. No significant variations were noted for most of the evaluated parameters among the three different groups at the conclusion of the trial compared to baseline (Table 1). However, a significant reduction in TC levels was observed for all the Placebo and intervention groups (*p* < 0.05, T90 vs. T0) as well as a significant decrease in FBG levels (*p* < 0.05, T90 vs. T0 for Placebo group; *p* < 0.01 for both *Cinchona* and Gengricin^®^ groups).

#### 2.2.2. Nutritional and Satiety Assessment

As shown in Table 2, no significant differences were reported in anthropometric and body composition parameters at the baseline among the three groups. After the three-month intervention period (T90), significant modifications were observed in all groups of the present study, compared to the baseline measurements (T0). Specifically, a significant reduction in body weight (−5.4%, *p <* 0.05 vs. T0) was reported for the Placebo group, while *Cinchona* and Gengricin^®^ exhibited the higher decrease in BW compared to baseline (−7.8%, *p <* 0.05, and −11.4%, *p <* 0.0001, respectively). Although BMI levels closely correlate with variations in BW, a significant decrease was exclusively achieved in the groups of individuals supplemented with the multi-component nutraceutical formulation (*p* < 0.05 vs. T0). As expected, the same trend of reduction was observed for waist (Placebo group: −4.1%, not significant; *Cinchona* group: −5.9%, *p* < 0.05; Gengricin^®^ group: −7.7%, < 0.05) and hip circumference values (Placebo group: −3.5%, *p* < 0.05; *Cinchona* group: −5.5%, *p* < 0.0001; Gengricin^®^: −7.5%, *p* < 0.0001) compared to baseline. Nonetheless, throughout the study period, all three groups reported an improvement in body composition, especially in terms of FM%, with the Placebo group showing a decrease of −5.1%, which was not statistically significant, while the Gengricin^®^ group demonstrated a more substantial reduction of −18.6% in FM (*p* < 0.001 vs. T0, *p* < 0.05 vs. Placebo T90).

Moreover, the results from satiety questionnaires at T90 showed that individuals of the intervention groups reported both decreased food craving compared to the Placebo group (scores: 4.3 ± 0.4, 2.7 ± 0.3, and 2.2 ± 0.3, respectively, for the Placebo, *Cinchona*, and Gengricin^®^ groups; *p* < 0.0001, Figure 2) and higher satiety at 4 h after meals (scores: 1.4 ± 0.2, 3.7 ± 0.5, and 4.2 ± 0.4, respectively, for the Placebo, *Cinchona*, and Gengricin^®^ groups; *p* < 0.0001). This suggests that supplementation with *Cinchona*, and to a greater extent with Gengricin^®^, actually contributed to prolonged satiety over the diet regimen period.

#### 2.2.3. Evaluation of Serum CCK and Ghrelin Levels

To assess the impact of nutraceutical formulations on EEC release, we herein measured the serum levels of CCK and ghrelin in all intervention and Placebo groups at two different time points, i.e., at the baseline (T0) and after 3 months (T90). Prior research has identified the expression of bitter taste receptors in both gastric and intestinal EECs. Gastric EECs are known to secrete ghrelin, whereas intestinal EECs predominantly secrete CCK [27]. Taking into account the delivery method of the nutraceutical formulations via gastro-resistant capsules, we hypothesized a more pronounced activation of intestinal EECs as compared to their gastric counterparts. As shown in Table 3, the hypocaloric diet led to a significant reduction in CCK levels at T90 in the Placebo group (−14.6%, *p <* 0.01), a variation associated with increased hunger. Conversely, in both the groups receiving the nutraceutical supplementation, there was no statistically significant reduction in CCK levels, suggesting that the nutraceuticals effectively stimulated intestinal EECs to release CCK. Notably, although not statistically significant, a higher increase in CCK release was observed in the Gengricin^®^ group compared to baseline (+8.9%, *p* = 0.132). Additionally, ghrelin levels remained unchanged in both the Placebo and treatment groups at the end of the study period, indicating no stimulation of gastric EECs.

## 3. Discussion

Obesity is a globally prevalent chronic disease that increases the risk factors for numerous pathological conditions, such as cardiovascular diseases, hypertension, infertility, diabetes mellitus, and dyslipidemia [28]. Previous studies have shown that a weight loss of 5–10% in subjects who are obese and affected by other comorbidities can significantly improve health and counteract mortality from many obesity-related diseases [29]. Recent years have seen a transformative understanding of the gastrointestinal tract, from being merely a site of nutrient digestion and absorption to being acknowledged as the body’s most extensive endocrine system [30]. It is known that more than 30 peptides are released from enteroendocrine cells in the gastrointestinal mucosa. These hormones interact with tissues both inside and outside the gut, significantly contributing to the regulation of metabolic balance [31]. Among these, ghrelin is mainly released by enteroendocrine cells of the stomach during fasting and it regulates food intake with a gastric acceleration. In contrast, CCK, glucagon-like peptide-1 (GLP-1), glucose-dependent insulinotropic polypeptide (GIP), and peptide YY (PPY) are predominantly released after a meal and interact to mediate intestinal feedback to reduce postprandial glycemic variations and to accelerate gastric emptying [32]. Currently, the pharmaceutical industry, recognizing the pleiotropic effects of gastrointestinal hormones in regulating metabolic homeostasis, is developing various synthetic drugs, especially GLP-1 receptor agonists and dual GLP-1/GIP agonists, to improve the management of obesity and related metabolic diseases [33]. Nevertheless, this approach often faces challenges such as side effects, costs, and suboptimal efficacy [34]. Based on this consideration, natural alternatives that can counteract or prevent obesity represent an alternative that shows significant promise.

Recent studies using preclinical and clinical models support the presence of a functional link between TAS2Rs in the gastrointestinal tract and the secretion of gastrointestinal hormones. This evidence indicates that the stimulation of TAS2Rs by bitter-tasting substances can be exploited to modulate gastrointestinal motility and energy intake in humans [15]. Modulating hormone secretion in the gastrointestinal tract using low or no-calorie compounds, such as bitter-tasting substances, would thus be advantageous compared to synthetic drugs. In this context, it is well known that enteroendocrine cells express a variety of bitter taste receptors [35]. The genomic organization of TAS2R gene clusters is notably coupled and these cells can have receptors for multiple bitter compounds, suggesting a broad sensitivity to bitterness [36], which can influence digestive processes and hormonal responses [37].

Based on these observations and considering that each receptor possesses unique specificity and affinity for different classes of bitter compounds, we herein aimed to evaluate the beneficial effects on satiety control and weight management exerted by a nutraceutical formulation containing a synergistic combination of different plant matrices appropriately designed to simultaneously target and stimulate these receptors. Specifically, as reported in Figure 1, the results obtained from an in vitro experiment performed on Caco2 cells showed a 15% increase in CCK release following incubation with the abovementioned nutraceutical formulation, named Gengricin^®^, compared to *Cinchona* alone (*p* < 0.05). This result supports the potential synergistic effect of the multi-component nutraceutical in increasing the activation of various bitter taste receptors. This synergism could be attributed to the combined phytochemical properties of the bioactive compounds, which may interact to more effectively stimulate a broader spectrum of bitter taste receptors [22,25]. This interaction potentially amplifies the overall stimulus on CCK-secreting cells, resulting in a more pronounced cellular response. Overall, these findings provide a compelling argument for the enhanced biologic activity of multi-component formulations compared to single ingredients, highlighting the importance of component synergies in nutraceutical research and development.

To corroborate these findings, we then moved to test the efficacy of the above-mentioned nutraceutical products in terms of effects on satiety, weight loss, and body composition change through a clinical trial conducted in subjects who were overweight–obese following a hypocaloric diet. In regards to the evaluation of biochemical parameters (Table 1), both the Placebo and intervention groups exhibited a significant reduction in TC and FBG from baseline, with a more pronounced effect observed in the groups of patients supplemented with the two nutraceutical formulations (*p* < 0.01 for both *Cinchona* and Gengricin^®^ groups). Additionally, the Gengricin^®^ group showed a slight decrease in TG levels compared to baseline (*p* < 0.05). Although the variations in the glucometabolic profile did not represent the primary endpoint of this study, the observed improvements in the lipid profile can be justified by the ameliorated metabolic condition achieved after weight loss [38].

In this regard, as detailed in Table 2, all study groups exhibited significant variations in BW values compared to baseline. Nevertheless, these changes were more pronounced in the Gengricin^®^ group compared to the *Cinchona* group (−11.4% vs. −7.8% from baseline, respectively), therefore emphasizing the additional effect of the multi-component nutraceutical formulation on weight loss. Notably, significant improvements in body composition were also observed in both the Placebo and intervention groups, as evidenced by the reductions in %FM and increases in %FFM, with the Gengricin^®^ group standing out as the more effective intervention compared to the *Cinchona* group (%FM: −18.6% vs. −13.1% and %FFM: +9.3% vs. +6.2% for Gengricin^®^ and *Cinchona* groups, respectively). Altogether, this evidence suggests that the multi-component nutraceutical formulation not only facilitates more substantial weight loss but also positively influences body composition, demonstrating its potential as a beneficial intervention in weight management strategies. Moreover, the differential impacts observed between the groups highlight the importance of formulation composition in achieving targeted health outcomes. Noteworthily, as reported in Figure 2, the results derived from satiety questionnaires at T90 indicate that the intervention groups, especially those supplemented with Gengricin^®^, experienced a significant reduction in food craving compared to the Placebo group. In this regard, the scores for the Placebo, *Cinchona*, and Gengricin^®^ groups were 4.3 ± 0.4, 2.7 ± 0.3, and 2.2 ± 0.3, respectively, with a statistically significant difference (*p* < 0.0001, Figure 2). Equally important is the observation regarding satiety levels 4 h post-meal, as *the Cinchona* and Gengricin^®^ groups reported higher satiety scores (3.7 ± 0.5 and 4.2 ± 0.4, respectively) compared to those in the Placebo group (1.4 ± 0.2), with these differences also being statistically significant (*p* < 0.0001). This result suggests that the supplementation with *Cinchona*, and more efficiently with Gengricin^®^, enhances the sensation of fullness post-consumption, contributing to prolonged satiety over the diet regimen period. Therefore, the supplementation with natural compounds like *Cinchona* and Gengricin^®^ could lead to an improved regulation in food intake, thereby playing a crucial role in dietary interventions.

In accordance with previous data, a pivotal aspect that emerged from this study is the observation of serum CCK response in the intervention groups (Table 3). In this regard, both the *Cinchona* and Gengricin^®^ groups demonstrated stable CCK levels, with a slight increase in serum CCK levels observed for the Gengricin^®^ group (+8.9%, *p* = 0.132), in contrast to typical decreases observed in conventional weight loss approaches. This conservative effect on CCK levels strongly suggests the beneficial potential of the tested nutraceutical formulations in mitigating the hunger pangs often associated with calorie-restricted diets, thereby enhancing the effectiveness of such natural weight management approaches [39]. The absence of a significant decrease in CCK levels within these groups is likely due to the bitter compounds occurring in the nutraceutical formulations. As supported by the in vitro study shown above, these compounds were able to activate TAS2Rs, thus stimulating CCK release. Moreover, the differential impact observed between the *Cinchona* and Gengricin groups could be further explored. While both groups showed efficacy in maintaining CCK levels, the nuanced differences in their formulations may account for the different responses in CCK release. This suggests a potential area for future research to analyze the specific components responsible for these effects and to optimize nutraceutical formulations for the targeted regulation of gut hormones.

The results obtained in the present study support the available scientific evidence on supplementation with nutraceutical formulations containing bitter compounds. In particular, an in vivo study published by Janssen et al. demonstrated that the intragastric administration of a mixture of TAS2R agonists, such as phenylthiocarbamide (PTC), denatonium benzoate (DB), quinine, and D-[-] salicin, can lead to an increase in plasma ghrelin levels. This was linked to an initial increase in food consumption in the first 30 min, followed by a sustained suppression of food intake over the following 4 h [13]. Moreover, a clinical trial published by Andreozzi and colleagues on healthy young participants showed that the oral administration of quinine in a gastro-resistant form led to an increase in CCK plasma concentrations and a decrease in energy expenditure during an ad libitum meal. In addition, a direct correlation between the degree of suppression of energy intake in response to quinine and the sensitivity of the subjects to the bitter taste of phenylthiocarbamide (PTC) was found [40]. More specifically, alkaloids from *Cinchona* bark were previously demonstrated to be a viable alternative in clinical practice for obesity treatment and its related dysmetabolic diseases [19,41,42]. In this regard, a very recent study has focused on the interaction between the quinoline alkaloids present in *Cinchona* bark and the bitter taste receptors TAS2Rs at the duodenal level [19]. In the aforementioned study, adults who were overweight/obese were treated for 60 days with a hypocaloric diet and either a nutraceutical supplementation based on *Cinchona* bark (treatment group) or a Placebo. For the treatment group, consisting of 32 adults, the observed plasma CCK levels were 14.21 ± 3.4 pg/mL at baseline (T0) and slightly decreased to 13.26 ± 1.3 pg/mL at the end of the study (T2). In contrast, the Placebo group of 27 adults showed a significant decrease in plasma CCK levels from 14.11 ± 2.21 pg/mL at T0 to 5.66 ± 1.10 pg/mL at T2, marking a substantial decrease of about 40%. This reduction in the Placebo group was associated with increased hunger typically observed in adults on hypocaloric diets. However, the treatment group did not experience a statistically significant reduction in CCK levels, suggesting that the nutraceutical supplementation was effective in stimulating intestinal EECs to release CCK, thereby maintaining satiety levels [19]. Noteworthily, results obtained from animal-based studies further supported the anti-obesity potential of *Cinchona*, as its supplementation showed significant effects in suppressing adipogenesis through the downregulation of WNT signaling pathways and galanin-mediated adipogenesis (orexigenic neuropeptide), concurrently with a reduction in inflammation through the repression of toll-like-receptor-2- (TLR2-) and TLR4-mediated signaling pathways in adipose tissue [41].

Additionally, numerous pieces of in vivo evidence highlighted the potential role of various bitter-tasting substances of a natural origin for the gastrointestinal modulation of hormone secretion through interaction with TAS2Rs. In particular, the oral administration of secoiridoid glycosides from the root of Gentian (*Gentiana lutea*), such as gentiopicrin [43], was shown to promote an increase in plasma CCK and GLP-1 concentrations, leading to a reduction in energy intake during meals in healthy subjects [43,44]. Sesquiterpene lactones, characterized by their strong bitter taste and present in the roots and leaves of various *Chicory* species, were also demonstrated to reduce appetite and subsequent caloric intake during meals. A 5-week pilot dietary intervention study conducted on mice revealed that various extracts of *Chicory* root, rich in lactucin and lactucopicrin, were highly effective in appetite control. This was evidenced by substantial alterations in the release of satiety hormones and changes in the composition of the intestinal microbiota. In particular, a significant increase in the plasma concentration of the gastrointestinal hormones CCK and GLP-1 was observed [45]. Altogether, these findings highlight the potential of natural nutraceutical strategies based on the combination of plant matrices rich in bitter compounds to modulate the release of gut hormones for effective appetite control and weight management.

## 4. Materials and Methods

### 4.1. Cell Culture Experiments

CaCo-2 cells were grown in Dulbecco’s modified Eagle’s medium (DMEM) (Cat. No. 41965-039, Gibco, Thermo Fisher Scientific, England, UK) supplemented with GlutaMAX (Cat. No. 35050-061, Gibco, Thermo Fisher Scientific, England, UK), 10% fetal bovine serum FBS (Cat. No. 10270, Gibco Thermo Fisher Scientific, England, UK), and Penicillin–Streptomycin (10,000 U/mL) (Cat. No. 15070-063, Gibco, England, Thermo Fisher Scientific). In cell experiments, the culture medium was replaced with a serum-free medium (SFM) (Gibco, Thermo Fisher Scientific, UK) containing either a *Cinchona* bark extract alone (named *Cinchona* group) or a multi-component nutraceutical formulation (*Cinchona* bark, *Chicory* root, and *Gentian* root in a 1:1:1 ratio), referred to as the Gengricin^®^ group. Following a 6 h treatment period, 200 μL of tissue supernatants was collected and transferred to ELISA NUNC Maxisorb plates. The plates were incubated for 16 h at 4 °C. Subsequently, the CCK levels were quantified using the Human CCK ELISA KIT (Sigma, Milan, Italy).

### 4.2. Randomized Controlled Trial (RCT)

#### 4.2.1. Study Population and Design

This study was designed as a three-month, three-arm, double-blind, randomized, Placebo-controlled trial. A total of 120 participants were recruited from the Clinic of the Departmental program “Diet therapy in transplantation and chronic renal failure”, School of Medicine and Surgery of the “Federico II” University of Naples, and from the “Studio Brofferio” and “Studi Generali Roma” medical offices in Rome, following a protocol approved by the Ethical Committee of the University of Naples Federico II. Criteria for inclusion encompassed both men and women over 18 years of age, with either an overweight status (BMI between 25 and 29.9) or obesity (BMI > 30), including patients with type 2 diabetes treated solely with a diet. Exclusion criteria included those involved in other clinical studies, inconsistent supplement intake, weight change over 3 kg in the last two months, a current or recent (within 5 years) cancer diagnosis, chronic metabolic and inflammatory diseases, prior diabetes treated with insulin or oral hypoglycemic agents, use of weight loss drugs, history of bariatric surgery, or hormonal therapies. This study protocol was approved by the Ethical Committee of the Federico II University Medical School of Naples—A.O.R.N. Cardarelli (EC approval code: 204/2023); all patients gave written informed consent. The protocol was registered with the ISRCTN13055163 number. Prior to enrollment, patients underwent a preliminary medical examination and provided informed consent after being briefed about the study’s purpose, objectives, and safety. This study comprised two phases: an initial evaluation (T0) involving biochemical and anthropometric measurements, a quantification of serum CCK and ghrelin levels, a body composition analysis, and questionnaires to gauge physical activity, dietary habits, and psychophysical well-being. Moreover, satiety was measured using visual analog scales (VASs) and a questionnaire that has been previously verified for its effectiveness [46]. The second phase consisted of a final assessment post the 90-day intervention period. During this study, the emergence of any exclusion criteria led to an immediate termination of the participant’s involvement in the trial. A total of 6 patients were found to be ineligible for this study. Ultimately, 114 patients received the allocated intervention, and 92 successfully completed this study (Figure 3).

#### 4.2.2. Food Supplementation

The participants were treated with a hypocaloric diet supplemented with one of three different treatments, administered as gastro-resistant capsules twice daily, one hour before main meals. The hypocaloric diet plan included a 40% caloric restriction of the total energy requirement [19,47] and the administration of a dietary supplement or Placebo. Resting energy expenditure was calculated using the Fredrix equation, adjusted for total energy expenditure. The *Cinchona Succirubra* bark powder (containing quinine at 22.27 mg/g, quinidine at 1.23 mg/g, cinchonine at 0.45 mg/g, and cinchonidine at 21.8 mg/g), *Gentian* L. root powder (containing gentiopicrin at 20 mg/g), and *Cichorium intybus* L. root powder (containing lactucin at 0.35 mg/g) were all provided by Farmalabor srl (Puglia, Italy). The supplement was provided in 60-capsule packs for the initial four weeks, with subsequent control visits and additional capsule supplies at 30 and 60 days.

The group assignments were as follows: Group 1 (Placebo group) received a Placebo containing 600 mg of maltodextrin twice/day; Group 2 (*Cinchona* group) received 600 mg of *Cinchona* bark twice/day; Group 3 (Gengricin^®^ group) received 600 mg of a mixture consisting of *Cinchona* bark, *Chicory* root, and *Gentian* root in a 1:1:1 ratio. The list of random numbers was generated by an investigator with no clinical involvement in this study. Patients, physicians, laboratory technicians, and trial personnel (data analysts and statisticians) were blinded to the treatment assignment.

#### 4.2.3. Assessments

In this study, participants’ nutritional and health status were assessed using a range of anthropometric and biochemical parameters. Initial evaluations involved measuring weight (using Seca GmbH & Co. KG equipment, Hamburg, Germany), height (via a precision wall-mounted stadiometer), body mass index (BMI), and circumferences of the waist (WCs) and hip (HCs). Further, a detailed body composition analysis was conducted using a bioelectrical impedance analysis (BIA) with a tetrapolar RJL 101 device (Akern SRL, Florence, Italy), focusing on total body water (TBW), fat mass (FM), and fat-free mass (FFM). Specifically, the primary endpoints measured were changes in BW and BMI. Secondary endpoints included changes in FM and FFM. Concurrently, all subjects underwent comprehensive physical examinations, including an evaluation of their medical history and vital signs. Adherence to the dietary plan was evaluated by recording dietary intake at the outset and monthly thereafter, up to this study’s conclusion, utilizing validated food frequency questionnaires [48]. To determine adherence to the physical activity regimen, participants were required to complete a physical activity questionnaire [49]. These questionnaires demonstrated that participants from both cohorts maintained regular engagement in physical activity during the entire course of the treatment period. The assessment of satiety levels was conducted employing visual analog scales (VASs). Furthermore, adherence to nutraceutical supplementation was monitored through a daily questionnaire, where participants reported the timing of supplement intake and any adverse events experienced. Concurrently, the count of remaining capsules at this study’s end was recorded to further assess compliance. Standardized laboratory tests were also performed, involving fasting blood samples collected at specified intervals (weeks 0, 4, 8, and 12) using EDTA tubes, followed by centrifugation and storage at −80 °C for a subsequent analysis. Key biochemical parameters, including fasting blood glucose (FBG); total, HDL, and LDL cholesterol; triglycerides; creatinine; and liver enzymes (ALT and AST), were regularly monitored using kits from Diacron International. Participants were advised to avoid alcohol and strenuous physical activity before these tests to ensure accuracy.

#### 4.2.4. Sample Size and Statistical Power

The sample size calculation was carried out using MedCalc software, version 22.021. The primary metric for determining the sample size was the decrease in body weight in the intervention groups. An anticipated 25% difference was projected between the treatment and Placebo group. The power and significance levels were established at 0.80 and 0.05, respectively. Based on these parameters, the required sample size was 30 participants for each group.

### 4.3. Quantification of Serum CCK and Ghrelin Levels

Levels of serum CCK and ghrelin were determined using the Dot-blot method. This involved placing five microliters of serum samples onto a nitrocellulose membrane, followed by processing using Western blotting techniques. The detection of the two enteroendocrine hormones was carried out using specific anti-cholecystokinin and anti-ghrelin antibodies (Santa Cruz, CA, USA), with a 1:250 dilution ratio.

### 4.4. Statistical Analysis

Results are presented as means ± standard deviation (St.Dev). Data were analyzed with a one-way ANOVA test followed by a Tukey–Kramer post hoc test using GraphPad Prism 8.4.3 software. A two-way ANOVA test followed by a Tukey–Kramer post hoc test was used to analyze changes in VAS scores of satiety and hunger. Significance was accepted at the 5% level.

## 5. Conclusions and Future Perspectives

Overall, the comprehensive investigation of how bitter compounds from *Cinchona*, *Gentian*, and *Chicory* synergistically interact with TAS2Rs provides valuable insights into their potential as plant matrices for the formulation of innovative nutraceutical products. By targeting a spectrum of TAS2Rs, these compounds hold promise in the development of nutraceuticals that harness the health benefits associated with the activation of specific bitter receptors, such as appetite control and body weight reduction. Nonetheless, the results from the clinical trial highlight the efficacy of a complex nutraceutical formulation combining these vegetable matrices as a useful food supplement, or even a potential alternative, to conventional pharmacotherapies for weight loss and body composition management. Further research should explore the long-term impacts of these nutraceutical compounds and their potential interactions with other medications and dietary components, thus contributing to the field of personalized nutrition. Additionally, studying the molecular mechanisms behind the specific activation of TAS2Rs exerted by these substances could provide deeper insights into their therapeutic potential. Lastly, extending this research to include a wider range of metabolic and obesity-related disorders may offer a more comprehensive understanding of the role these nutraceuticals play in the broader context of health and disease management.

## 6. Patents

Patent Number: 102021000005429. Patent Title: “Food supplement to promote body weight loss”. Inventors: Ettore Novellino and Gian Carlo Tenore. Publication Date: 21 March 2023.

## Figures and Tables

**Figure 1 ijms-25-02596-f001:**
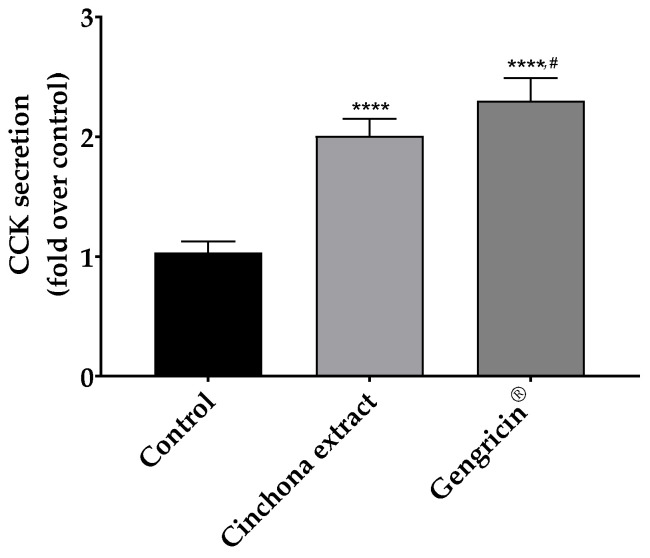
Cholecystokin (CCK) secretion in CaCo-2 cells upon stimulation with *Cinchona* or Gengricin^®^ extract. Values indicate the fold increase in hormone secretion compared to untreated cells (control). Fold increases are reported as mean ± St. Dev of 5 replicates. Data were analyzed with a one-way ANOVA followed by Tukey’s multiple comparison post hoc test; ****, *p* < 0.0001 compared to control cells, #, *p* < 0.05 compared to *Cinchona* extract.

**Figure 2 ijms-25-02596-f002:**
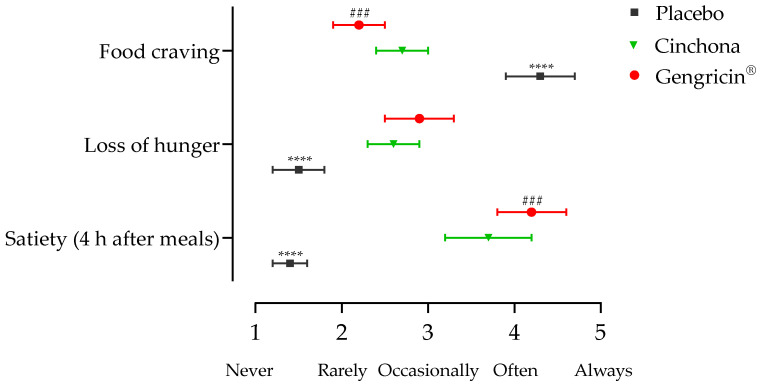
Changes in VAS score of satiety and hunger after 3-month intervention period. Values are reported as mean ± St. Dev. Data were analyzed with a two-way ANOVA followed by Tukey’s multiple comparison post hoc test; ****, *p* < 0.0001 compared to *Cinchona* and Gengricin^®^ groups, ###, *p* < 0.001 compared to *Cinchona* group.

**Figure 3 ijms-25-02596-f003:**
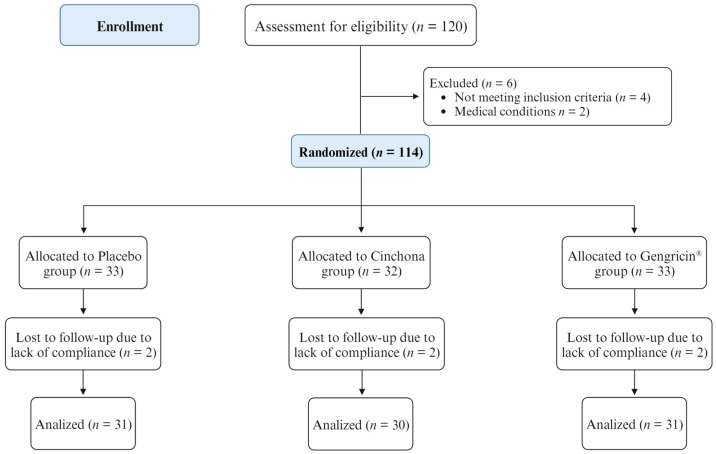
Study Consolidated Standards of Reporting Trials (CONSORT) flow diagram.

**Table 1 ijms-25-02596-t001:** General data and biochemical parameters of Placebo and intervention groups at baseline and after 3 months.

Parameters	Placebo Group (*n* = 31)		*Cinchona* Group (*n* = 30)		Gengricin^®^ Group (*n* = 31)	
	T0	T90	T0	T90	T0	T90
Male, n° (%)	17 (54.8)	-	12 (40)	-	16 (51.6)	-
Female, n° (%)	14 (54.2)	-	18 (60)	-	15 (48.4)	-
Age (years)	43 ± 6.2	-	47 ± 6.8	-	45 ± 5.9	-
FBG (mg/dL)	95.4 ± 4.6	91.2 ± 5.4 *	94.4 ± 66	89.4 ± 7.4 **	96.1 ± 8.3	90.4 ± 4.5 **
HDL-C (mg/dL)	42.2 ± 1.8	41.0 ± 1.3	44.2 ± 4.7	44.0 ± 3.9	45.7 ± 3.5	46.6 ± 4.2
LDL-C (mg/dL)	103.4 ± 8.8	102.7 ± 6.5	109.5 ± 9.8	108.8 ± 11.1	114.5 ± 15.9	112.6 ± 13.8
TC (mg/dL)	188.4 ± 14.1	177.8 ± 17.9 *	187.5 ± 14.2	176.5 ± 12.5 *	190.5 ± 12.4	181.6 ± 14.3 *
TGs (mg/dL)	134.3 ± 22.1	130.6 ± 21.9	122.7 ± 17.5	118.5 ± 18.5	137.5 ± 22.1	123.6 ± 13.2 *
AST (UI/L)	25.6 ± 7.2	24.7 ± 6.2	21.5 ± 6.3	20.1 ± 5.6	23.5 ± 3.6	24.3 ± 2.8
ALT (UI/L)	18.9 ± 4.3	19.0 ± 2.3	19.3 ± 3.7	18.4 ± 3.6	20.8 ± 5.6	21.4 ± 5.5
Cre (mg/dL)	0.9 ± 0.1	0.9 ± 0.2	1.1 ± 0.1	1.2 ± 0.2	1.2 ± 0.4	1.1 ± 0.2

Data are expressed as mean ± standard deviation. * *p* < 0.05, ** *p* < 0.01, T90 vs. T0. Abbreviations—AST: aspartate aminotransferase; ALT: alanine aminotransferase; Cre: creatinine; FPG: fasting blood glucose; HDL-C: high-density lipoprotein cholesterol; LDL-C: low-density lipoprotein cholesterol; TC: total cholesterol; TGs: triglycerides.

**Table 2 ijms-25-02596-t002:** Anthropometric and body composition parameters of Placebo and intervention groups at baseline and after 3 months.

Parameters	Placebo Group (*n* = 31)		*Cinchona* Group (*n* = 30)		Gengricin^®^ Group (*n* = 31)	
	T0	T90	T0	T90	T0	T90
Weight (kg)	93.4 ± 7.0	88.4 ± 8.3 *	94.1 ± 8.2	86.8 ± 7.3 **	95.4 ± 9.9	84.5 ± 8.4 ****
BMI (kg/m^2^)	32.3 ± 6.2	30.6 ± 2.9	32.6 ± 5.7	30.0 ± 6.4	33.0 ± 6.7	29.2 ± 5.8 *
WC (cm)	103.1 ± 7.5	98.9 ± 6.6	102.8 ± 6.8	96.7 ± 7.9 *	103.5 ± 7.8	95.5 ± 8.6 **
HC (cm)	123.2 ± 4.5	118.9 ± 6.5 *	122.4 ± 5.3	115.7 ± 4.8 ****	124.2 ± 4.5	114.9 ± 7.2 ****
TBW (%)	51.2 ± 3.5	52.9 ± 4.6	51.4 ± 5.3	52.7 ± 5.8	49.8 ± 6.5	53.4 ± 5.2
FM (%)	33.2 ± 3.4	31.5 ± 5.7	32.0 ± 4.8	27.8 ± 5.3 *	33.4 ± 6.2	27.2 ± 7.4 ***^,#^
FFM (%)	66.8 ± 3.4	68.5 ± 5.7	68.0 ± 4.8	72.2 ± 5.3 *	66.6 ± 6.2	72.8 ± 7.4 ***^,#^

Data are expressed as mean ± standard deviation. * *p* < 0.05, ** *p* < 0.01, *** *p* < 0.001, **** *p* < 0.0001, T90 vs. T0; ^#^ *p* < 0.05 vs. Placebo T90. Abbreviations: BMI, body mass index; FFM, fat-free mass; FM, fat mass; HC, hip circumference; TBW, total body water; WC, waist circumference.

**Table 3 ijms-25-02596-t003:** Enteroendocrine hormone levels of Placebo and intervention groups at baseline and after 3 months.

	Placebo Group (*n* = 31)		*Cinchona* Group (*n* = 30)		Gengricin^®^ Group (*n* = 31)	
	T0	T90	T0	T90	T0	T90
CCK (pg/mL)	14.4 ± 2.9	12.3 ± 3.8 **	14.1 ± 2.2	13.8 ± 1.3	14.5 ± 1.9	15.8 ± 2.4
Ghrelin (ng/mL)	7.2 ± 1.1	7.1 ± 1.3	7.6 ± 2.5	7.5 ± 1. 4	7.2 ± 1.3	7.3 ± 1.2

Data are expressed as mean ± standard deviation. ** *p* < 0.01, T90 vs. T0.

## Data Availability

The data used to support the findings of this study are included within the article.

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
