# Peer review of "Gengricin®: A Nutraceutical Formulation for Appetite Control and Therapeutic Weight Management in Adults Who Are Overweight/Obese"

_ijms, 2024, doi:10.3390/ijms25052596_

Round 1

Reviewer 1 Report

Comments and Suggestions for Authors

Dear authors

The idea of manuscript is very good and writting well.

Obesity represents a significant risk factor for numerous pathologies, including cardiovascular diseases, type 2 diabetes, and cancers. Currently, obesity has reached epidemic proportions worldwide, affecting both adults and children, resulting in a substantial financial burden on healthcare systems. Reduction of obesity can significantly improve health and counteract mortality from many obesity-related diseases. So, searching for nutraceuticals for reduction of obesity and studying the mechanism of action and the target sites which these nutraceuticals play on for reducing obesity are very important strategy.

The aim of the present study was to evaluate the effect of a nutraceutical formulation, Gengricin®, containing a combination of molecules appropriately designed to simultaneously target and stimulate TAS2R, which is useful for obesity management and satiety control.

The results of the study were logic and discussed well.

References of the manuscript were up to date, but some references need to be reviewed.

Reviewer 2 Report

Comments and Suggestions for Authors

The authors presented a manuscript entitled "Gengricin®: a Nutraceutical Formulation for Appetite Control and Therapeutic Weight Management in Overweight/Obese Adults". The study evaluated the effect of a nutraceutical formulation containing a combination of molecules appropriately designed to stimulate TAS2R receptors. The manuscript is well-written and recent references have been cited.

While the study provides valuable insights, there are a few aspects that can be improved:

1. The authors did not provide information about the molecular mechanism and the nature of TAS2R receptors. Please, add some information.

2. On what the basis was it determined that the dietary plan included a 40% calorie restriction? Please, define it or insert a reference to the relevant paragraph.

3. On what basis was the 600 mg dose selected?

4. Please provide a more specific statistical analysis. Was one-way ANOVA used?

5. Did the authors make any comparisons between placebo and intervention groups? (related to the data of the tables)

Round 2

Reviewer 2 Report

Comments and Suggestions for Authors

I accept the manuscript in present form.